# Economic segregation is associated with reduced concerns about economic inequality

Shai Davidai ⬤[1] ✉, Daniela Goya-Tocchetto ⬤[2] & M. Asher Lawson ⬤[3]

Economic segregation is the geographical separation of people with different economic means. In this paper, we employ an archival study of attitudes in regions with varying degrees of economic segregation and a series of experimental studies measuring reactions to hypothetical levels of segregation to examine how segregation affects concerns about inequality. Combining correlational and experimental methods and examining attitudes about economic inequality in both the United States and South Africa, we show that when individuals of different means are segregated from each other, people are less likely to engage in economic comparisons and are therefore less concerned by inequality. Moreover, we find that this is true even when people are exposed to (and are aware of) the same levels of inequality, suggesting that segregation in and of itself affects attitudes about inequality. Our findings highlight the importance of economic segregation in shaping public attitudes about organizational and societal economic inequality.

Social scientists have long sought to understand why people do not seem as concerned about economic inequality as one might expect them to be[1]. Whereas some researchers have examined this question by focusing on ideological differences[2–4], lay beliefs about inequality[5–7], and misperceptions about it[8,9], others have taken a more situational approach, emphasizing how exposure to inequality shapes concerns about it[10–17]. Yet, since highly unequal regions tend to be economically segregated[18], living in unequal areas may not be sufficient for increasing people's concern about inequality. In this paper, we suggest that economic segregation—the separation of people with different economic means—is a critical factor that shapes attitudes about inequality. Specifically, we predict that economic segregation reduces people's concerns about economic inequality.

Economic segregation shapes many aspects of our lives, affecting where we live, work, shop, pray, play, and go to school[19–22]. For instance, college students from families in the top 1% are more likely to meet similarly wealthy students at elite universities than they are to meet students from the entire bottom half of the income distribution[21]. And, while people of different financial means are obviously aware of each other's existence, economic segregation dampens cross-class interactions and fosters class isolation[19,20,23]. Consequently, as wealthier individuals segregate into homogeneously affluent neighborhoods, they less frequently interact with middle- and lower-class others and are thus less frequently exposed to situations where the juxtaposition of wealth and poverty is salient[24,25].

Consider, as an illustrative example from visual perception, the famous Mach Bands optical illusion, in which the contrast between bands of different shades of gray is made salient by their juxtaposition[26]. Just as the contrast between different shades of color is more visually salient when they are directly adjacent to each other than physically separated, we argue that the economic contrast between different levels of wealth is more psychologically salient when individuals of different means are functionally close to each other than when they are segregated. And, just as comparing different shades of color is visually more difficult when they are physically separated, we suggest that comparing different levels of wealth is more psychologically difficult when economic segregation is high. Put differently, by rendering the juxtaposition between people of different means less apparent, economic segregation creates an obstacle to economic comparisons. Thus, even if people know about the existence of inequality, we argue that segregation limits their ability to fully compare differences in finances between people of different means.

Drawing from research on basic cognitive and social processes, we suggest that economic segregation decreases concerns about

[1]Columbia Business School, New York, NY, USA. [2]University at Buffalo SUNY, Buffalo, NY, USA. [3]INSEAD, Fontainebleau, France.
✉ e-mail: sd3311@columbia.edu

inequality by impeding economic comparisons. Salient economic reference groups make it easy to compare different levels of wealth to each other and thus shape support for redistributive policies. For instance, people are more supportive of redistributive policies when they personally know someone who has economic problems or are in contact with someone who is unemployed[27–30]. And, since making sense of inherently ambiguous concepts like 'financial success' requires comparing financially well-off individuals to less fortunate others[31,41], less frequent engagement in such comparisons can affect attitudes about inequality. Thus, we predict that the dampening effect of segregation on economic comparisons reduces concerns about inequality. Since economic segregation inherently reduces the contrast between wealth and poverty, we hypothesize that it makes comparisons less salient and economic disparities appear as less troubling.

Importantly, in contrast to past research[12–17,23,42], we examine the psychological effects of economic segregation even when people are exposed to the very same level of inequality. By sorting people of different economic means into separate groups that live in distinct areas and functionally segregated institutions, segregation reduces the available cues that people have about the actual level of inequality in society. And, since people evaluate societal inequality by sampling their immediate environments[10,35], this lack of cues of inequality in highly segregated areas may lead people to underestimate its full extent[12–17]. Yet, beyond focusing on such distorted perceptions, we propose that the effect of segregation on concerns about inequality occurs through an independent pathway: reduced economic comparisons. As is the case with the Mach Bands illusion, we suggest that caring about inequality requires more than mere exposure to different levels of wealth. Rather, we argue that the direct juxtaposition between people of different means is critical for shaping concerns about inequality.

Consider, for example, how the same level of inequality is experienced differently at different levels of economic segregation. Imagine someone who lives in a highly segregated area who passes a homeless person while driving back from an expensive meal versus someone who lives in an economically integrated area and who sees the homeless person lying on the sidewalk outside of the restaurant while they enjoy their expensive meal. Although both situations involve the same level of inequality (i.e., a wealthy person at a restaurant versus a homeless person living on the street), the juxtaposition of fine-dining and homelessness in an economically integrated area makes the contrast between financially better-off and worse-off individuals more salient and may therefore increase concerns about it. Thus, even when the level of inequality remains the same, segregation may obscure the juxtaposition between wealth and poverty, impeding economic comparisons and creating an impression of fairness. That is, even when people see identical economic disparities, we argue that economic segregation reduces concerns about inequality.

Using archival and experimental methods and drawing from lab and online research samples, we examine whether increases in economic segregation are related to reduced concerns about inequality. First, combining data capturing levels of economic segregation across the U.S. with over two decades of survey data from the World Values Survey, we examine whether living in more segregated regions is associated with lower concern about inequality. Following, a series of studies using real and hypothetical scenarios and employing between-participants and within-participants research designs examine the causal effect of actual and perceived segregation on concerns about inequality, testing whether it is (at least partially) due to reduced comparisons, and whether segregating people based on their socioeconomic status reduces concerns about inequality even when people know about the distribution of income. Thus, beyond affecting perceptions of inequality, we examine whether segregation reduces concerns about inequality through an additional pathway: reduced

comparisons. In all studies, we find that segregation is robustly related to reduced concerns about inequality and that this is true even when holding constant both the actual and the perceived gaps between people of different economic means.

## Results
### Study 1
We drew on two large archival datasets to examine whether actual economic segregation in the U.S. is associated with reduced concerns about inequality. To do so, we combined data measuring economic segregation across the U.S.[43] with over two decades of data regarding concern about inequality from the World Values Survey[44]. The economic segregation data contained three different measures of segregation and three different measures of inequality within 741 Commuter Zones across the US. The concern about inequality data was derived from 8,764 unique observations across 50 states from five waves of data collection (1995–2017) in the World Values Survey. Based on previous research, we focused on responses to a single-item measuring attitudes about inequality, asking respondents to indicate (on a 10-point scale) how much they believe that "Incomes should be made more equal"[1] versus "We need larger income differences as incentives"[10] (reverse scored, so that higher values reflect more concern about inequality).

Many possibilities for analyses arise from the complexity of the merged dataset, with at least three major axes for potential researcher degrees of freedom: (1) whether to use an overall measure of segregation, a measure of the segregation of high-income households from less affluent households, or a measure of the segregation of low-income households from more affluent households, (2) whether to use (as a control variable) an area's overall Gini index, the Gini index for the bottom 99%, or the share of income held by the 1%, and (3) whether to control for additional variables[43]. As a conservative test of our hypothesis, we estimated all potential models using the full 18 combinations of these parameters and tested the results when excluding data from one state at a time, producing a total of 918 estimates of the effect. In Table 1, we present two example models using basic specifications: one using the overall measure of segregation to predict attitudes about inequality while controlling for the overall Gini coefficient (Model 1) and one that also controls for population, income per capita, proportion of Black residents, racial segregation, and proportion of votes for the Democratic party in the 2020 Presidential election (Model 2). Following, we report a summary of all 918 specifications, examining the findings' robustness to different analytical choices across the full grid of parameters.

Providing evidence for the relationship between economic segregation and attitudes about inequality, we found that living in segregated areas is significantly associated with lower concern about inequality. A linear regression with robust standard errors (clustered within state and year) revealed a negative relationship between segregation and concerns about inequality both when controlling for the level of economic inequality (standardized coefficient: $\beta = -0.061$, $SE = 0.028$, $p = 0.029$; 95% CI[−0.116, −0.006]) and for a wider set of control variables (standardized coefficient: $\beta = -0.269$, $SE = 0.093$, $p = 0.004$; 95% CI[−0.451, −0.087]). Regardless of the actual level of economic inequality, the more people were geographically segregated from others with higher or lower incomes, the less they worried about such disparities.

We next examined the robustness of this effect across a comprehensive set of all potential specifications. Across the 918 specifications, we found an average effect size of $b = -0.150$, suggesting a modest and significant relationship between economic segregation and concern about inequality. Of these estimates, all (100%) were negative, and 98.1% were statistically significant (Fig. 1). In the Supplemental Materials we report additional exploratory analyses using an alternative approach for data aggregation that yielded very similar

**Table 1 | Results from two-sided multiple linear regression analyses predicting attitudes about inequality from economic segregation, Gini coefficient, Population Income per capita, Proportion of Black residents, Racial segregation, and Proportion of Democrat vote**

| | Model 1 | Model 2 |
|---|---|---|
| Economic segregation | | |
| *Coefficient [95% CI]* | −0.061 [−0.116, −0.006] | −0.269 [−0.451, −0.087] |
| *p* | 0.029 | 0.004 |
| Gini coefficient | | |
| *Coefficient [95% CI]* | −0.003 [−0.027, 0.022] | 0.075 [−0.008, 0.157] |
| *p* | 0.821 | 0.071 |
| Population | | |
| *Coefficient [95% CI]* | | 0.018 [−0.054, 0.089] |
| *p* | | 0.626 |
| Income per capita | | |
| *Coefficient [95% CI]* | | 0.131 [0.070, 0.192] |
| *p* | | <0.001 |
| Proportion of Black residents | | |
| *Coefficient [95% CI]* | | −0.058 [−0.127, 0.011] |
| *p* | | 0.097 |
| Racial segregation | | |
| *Coefficient [95% CI]* | | 0.103 [−0.006, 0.212] |
| *p* | | 0.064 |
| Proportion Democrats | | |
| *Coefficient [95% CI]* | | −0.031 [−0.055, −0.007] |
| *p* | | 0.012 |
| Constant | | |
| *Coefficient [95% CI]* | 0.000 [−0.144, 0.144] | −0.000 [−0.120, 0.120] |
| *p* | 0.999 | 0.997 |

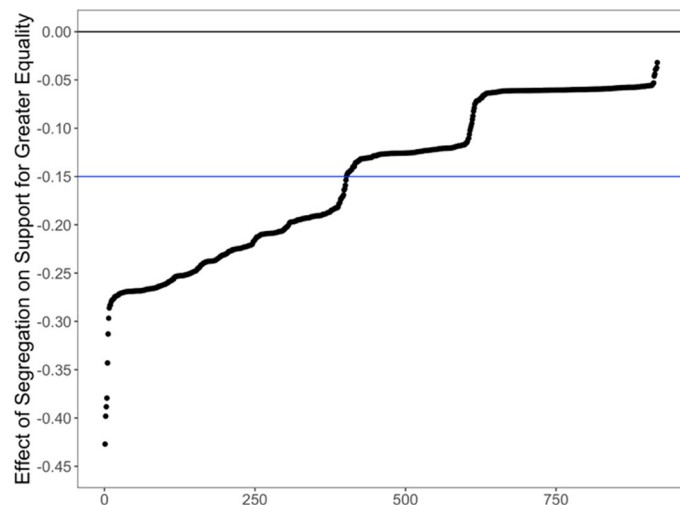

**Fig. 1 | Specification curve for the effect of economic segregation on attitudes about economic inequality across 918 specifications.** The horizontal blue line denotes the average effect size across specifications. The horizontal black line denotes a null effect (i.e., 0).

results (albeit with a somewhat smaller average effect size). Regardless of analytical choice, we consistently found that economic segregation is negatively associated with concern about inequality. Moreover, we report in the Supplementary Materials exploratory analyses of the effect of segregation across different levels of inequality. Although this effect appeared to be slightly attenuated at higher levels of inequality, the interaction effect varied substantially, with significant positive (28.0%) and negative (15.4%) effects across different specifications.

In Studies 2-5, we examined the causal effect of a hypothetical level of segregation on attitudes about inequality. To do so, participants imagined working or living in highly segregated places and indicated their concerns about inequality. We predicted that merely imagining being in a highly segregated place reduces concerns about economic disparities.

## Study 2a

Participants (N = 299) imagined working at one of two companies with different levels of economic segregation. They saw images of 80 employees and their respective salaries across four offices and were randomly assigned to one of two conditions, both of which involved the same level of inequality but differed in the segregation of wages (Table 2). In the Economic Segregation condition, participants saw a company where each office was relatively homogenous in terms of pay (i.e., 20 highest-paid employees worked at a different office from the second 20 highest-paid employees, who worked at a different office from the next 20 highly paid employees, and so on). In the Economic Integration condition, participants saw a company where each office employed some of the company's highest and lowest-paid employees (i.e., the contrast between higher and lower paid employees was apparent within each

office). Importantly, both conditions had equal levels of organizational inequality and only varied on whether the inequality occurred between (Economic Segregation) or within (Economic Integration) offices.

Participants saw one office at a time (in random order, with each office represented by 20 employees and their respective salaries) and indicated, as a manipulation check, the level of inequality in each office. Then, after seeing all four offices, participants indicated their beliefs about the overall level of inequality in the company: how unequal was the distribution of wages in the organization, how fair they perceived it to be, how concerned they were by it, and whether they saw it as a problem that merits the company's attention.

First, we examined the manipulation check. As intended, a two-tailed independent-sample t-test revealed that participants perceived less inequality within each office in the Economic Segregation condition ($M = 2.79$, $SD = 1.23$) than the Economic Integration condition ($M = 5.09$, $SD = 1.49$; $M_{diff} = 2.30$; two-tailed independent-sample $t$(269.89) = 14.26, $p < 0.001$, $d = 1.681$, 95% CI [1.98, 2.62]). Since this and all following studies focused on a set of specific, theoretically-based planned comparisons (rather than every possible comparison), no correction for multiple comparisons was required.

Next, we used a series of two-tailed t-tests to examine whether the hypothetical level of segregation affected concerns about the company's overall level of inequality. As predicted, participants saw the company as less unequal, more fair, and were less concerned by the overall wage disparity when it mainly occurred between offices (Economic Segregation) than within each office (Economic Integration) (Inequality: $M_{diff} = 0.72$, $t$(284.24) = 3.54, $p < 0.001$, $d = 0.414$, 95% CI [0.32, 1.13]; Fairness: $M_{diff} = -0.88$, two-tailed independent-sample $t$(287.55) = −4.01, $p < 0.001$, $d = 0.471$, 95% CI[−0.45, −0.22]; Concern: $M_{diff} = 1.11$, two-tailed independent-sample $t$(287.69) = 4.79, $p < 0.001$, $d = 0.562$, 95% CI [0.65, 1.56]). Participants also viewed inequality as a less pressing issue in the Economic Segregation condition than the Economic Integration condition (Inequality is a big issue: $M_{diff} = 1.04$, two-tailed independent-sample $t$(285.28) = 4.90, $p < 0.001$, $d = 0.574$, 95% CI [0.62, 1.45]; Need to decrease inequality: $M_{diff} = 0.59$, two-tailed independent-sample $t$(288) = 2.76, $p < 0.001$, $d = 0.325$, 95% CI[0.17, 1.00]). Thus, seeing inequality as mainly occurring between offices rather than within offices reduced concerns about it. Moreover, the fact that participants were exposed to the same level of inequality in both conditions reveals that segregating offices based on wages played a distinct role in reducing concerns about inequality.

**Table 2 | Distribution of wages (in thousands of dollars), office Gini coefficient, and overall company Gini coefficient in the Economic Segregation and Economic Integration conditions (Study 2a)**

| Condition | Office | Employees' annual wages (in thousands of dollars) | Office inequality (Gini) | Overall company inequality (Gini) |
|---|---|---|---|---|
| Economic segregation | 1 | 350, 350, 350, 340, 340, 315, 310,310, 300, 310, 310, 300, 300, 300, 300, 300, 300, 300 | .03 | 0.42 |
| | 2 | 138, 138, 138, 138, 134, 134, 134, 134, 127, 127, 127, 127, 110, 110, 110, 110 | .045 | |
| | 3 | 89, 89, 89, 89, 82, 82, 82, 82, 77, 77, 77, 77, 71, 71, 71, 71 | .046 | |
| | 4 | 34, 34, 34, 34, 32, 32, 32, 32, 29, 29, 27, 27, 27, 25 | .054 | |
| Economic integration | 1 | 340, 320, 140, 120, 100, 100, 84, 80, 80, 71, 22, 22, 20, 20, 20, 20 | .42 | 0.42 |
| | 2 | 340, 320, 140, 120, 100, 100, 84, 80, 80, 71, 22, 22, 20, 20, 20, 20 | .42 | |
| | 3 | 340, 320, 140, 120, 100, 100, 84, 80, 80, 71, 22, 22, 20, 20, 20, 20 | .42 | |
| | 4 | 340, 320, 140, 120, 100, 100, 84, 80, 80, 71, 22, 22, 20, 20, 20, 20 | .42 | |

To rule-out inferences about purchasing power, all four offices were said to be in the same region.

## Studies 2b and 2c

Using a hypothetical scenario, we found that people are less troubled by inequality when high- and low-income employees work in separate offices (i.e., economic segregation) than when they share the same office. Next, we replicated these findings with a conservative test of our hypothesis in which participants in a pre-registered online sample (2b; N = 100) and in a behavioral research lab (2c; N = 117) saw two potential groupings of a company's workforce that varied by economic segregation. Specifically, in a within-participant design, participants read about 20 employees of a company whose salaries varied between $27,000–$350,000 and saw, in counterbalanced order, two ways in which these employees can be grouped across four offices such that the wage disparity mostly occurred between or within offices. After seeing each potential grouping, participants indicated, as a manipulation check, the level of inequality in each office as well as their beliefs about the overall level of inequality in the company: how concerned they were by it, how fair they perceived it to be, and how equal or unequal it was.

We first examined whether the manipulation of segregation was successful. As intended, participants saw less inequality within each office when employees were segregated by wages ($M_{2b}$ = 2.41, $SD_{2b}$ = 1.47; $M_{2c}$ = 2.30, $SD_{2c}$ = 1.11) than not ($M_{2b}$ = 5.59, $SD_{2b}$ = 1.54; $M_{2c}$ = 5.60, $SD_{2c}$ = 1.51), matched-pairs t-tests $t(91)_{2b}$ = 16.05, $p_{2b}$ < 0.001, $d$ = 1.673, 95% CI[2.79, 3.57]; $t(110)_{2c}$ = 18.30, $p_{2c}$ < 0.001, $d$ = 1.737, 95% CI[2.94, 3.66].

Next, we examined judgments of the overall level of inequality. Unsurprisingly, since participants saw two groupings of the same 20 employees, we did not find a significant difference in perceptions of the overall distribution of wages in the company between condition, matched pairs t-tests $t(91)_{2b}$ = 0.89, $p_{2b}$ = 0.375, $d$ = 0.093, 95% CI[−0.19, 0.49]; $t(110)_{2c}$ = 0.97, $p_{2c}$ = 0.336, $d$ = 0.092, 95% CI[−0.12, 0.36]. Yet, despite not finding evidence of perceived differences in overall inequality, participants were less concerned about inequality ($M_{diff-2b}$ = −0.82, matched-pair $t(91)$ = 4.54, $p$ < 0.001, $d$ = 0.473, 95% CI[−1.17, −0.46]; $M_{diff-2c}$ = −0.97, matched-pair $t(110)$ = 5.00, $p$ < 0.001, $d$ = 0.474, 95% CI[−1.36, −0.59]) and judged it as significantly more fair ($M_{diff-2b}$ = 0.51, matched-pair $t(91)$ = 2.87, $p$ = 0.005, $d$ = 0.299, 95% CI[0.16, 0.86]; $M_{diff-2c}$ = 0.64, matched-pair $t(110)$ = 3.30, $p$ = 0.001, $d$ = 0.312, 95% CI[0.25, 1.03]) when it mostly occurred between offices than within offices. Thus, although participants saw the same 20 employees grouped in two different ways (i.e., a within-participant design), and despite noting that the overall level of inequality was the same, they were less concerned when it mainly occurred between offices (i.e., economic segregation) than within offices.

## Study 2d

Participants were less concerned about inequality in a hypothetical organization when it occurred between offices than within offices. This was found in both a pre-registered, online study (Study 2b) as well as in a direct replication conducted in a university research lab in the southeast of the U.S. (Study 2c). Study 2d examined whether such judgments of inequality are moderated by the perceived overall level of organizational inequality.

Participants (N = 602) imagined working at a company and were assigned to either an Economic Segregation condition (where inequality mostly occurred between offices, such that wages within each office were relatively homogenous) or an Economic Integration condition (where inequality mostly occurred within offices, such that every office employed both higher- and lower-paid employees). In addition, participants were assigned to one of four levels of inequality, varying the dispersion of wages across the company's entire workforce: relatively Low (Gini coefficient = 0.22), Medium (Gini coefficient = 0.32), High (Gini coefficient = 0.42), or Very High (Gini coefficient = 0.52). Importantly, within each level of inequality, participants in both

**Table 3 | Judgments of varying levels of inequality (Low, Medium, High, and Very High conditions) that occur within offices (Economic Integration conditions) or between offices (Economic Segregation conditions) (Study 2d)**

| Measure | Condition | Level of inequality | | | | Source | F Ratio | η² | p |
|---|---|---|---|---|---|---|---|---|---|
| | | Low | Medium | High | Very High | | | | |
| Perceived inequality | Integrated | 4.671 [4.281, 5.061] | 4.481 [4.028, 4.933] | 5.096 [4.73, 5.455] | 5.614 [5.224 6.004] | Segregation | 39.29 | 0.064 | 0.0001 |
| | Segregated | 3.770 [3.391, 4.150] | 3.681 [3.339, 4.023] | 3.908 [3.503, 4.312] | 5.013 [4.646, 5.380] | Level of inequality | 15.89 | 0.076 | 0.0001 |
| | | | | | | Interaction | 0.79 | 0.004 | 0.498 |
| Perceived fairness | Integrated | 4.114 [3.709, 4.519] | 4.192 [3.722, 4.662] | 3.289 [2.917, 3.661] | 2.657 [2.252, 3.062] | Segregation | 16.32 | 0.028 | 0.0001 |
| | Segregated | 4.595 [4.201, 4.989] | 4.330 [3.974, 4.685] | 4.185 [3.764, 4.605] | 3.481 [3.100, 3.862] | Level of inequality | 16.94 | 0.081 | 0.0001 |
| | | | | | | Interaction | 1.41 | 0.007 | 0.238 |
| Concern about inequality | Integrated | 3.957 [3.521, 4.393] | 3.885 [3.379, 4.391] | 5.024 [4.624, 5.425] | 5.471 [5.035, 5.908] | Segregation | 12.94 | 0.022 | 0.0003 |
| | Segregated | 3.527 [3.103, 3.951] | 3.692 [3.310, 4.075] | 4.092 [3.640, 4.545] | 4.785 [4.374, 5.195] | Level of inequality | 18.41 | 0.088 | 0.0001 |
| | | | | | | Interaction | 1.03 | 0.005 | 0.377 |
| Inequality is a big issue for workers. | Integrated | 4.429 [4.000, 4.857] | 4.327 [3.829, 4.824] | 5.241 [4.847, 5.635] | 5.771 [5.343, 6.200] | Segregation | 26.87 | 0.045 | 0.0001 |
| | Segregated | 3.689 [3.272, 4.106] | 3.956 [3.580, 4.332] | 4.138 [3.693, 4.583] | 4.810 [4.407, 5.214] | Level of inequality | 14.09 | 0.068 | 0.0001 |
| | | | | | | Interaction | 1.05 | 0.005 | 0.369 |
| Company should implement policies that decrease wage disparities | Integrated | 4.586 [4.165, 5.006] | 4.462 [3.973, 4.950] | 5.434 [5.047, 5.820] | 5.871 [5.451, 6.292] | Segregation | 8.80 | 0.015 | 0.0031 |
| | Segregated | 4.216 [3.807 4.625] | 4.637 [4.268, 5.002] | 4.615 [4.179, 5.052] | 5.101 [4.705, 5.497] | Level of inequality | 10.91 | 0.054 | 0.0001 |
| | | | | | | Interaction | 2.27 | 0.012 | 0.079 |

conditions saw the same distribution of wages but differed in whether the inequality occurred between or within offices.

As before, participants saw one office at a time (i.e., 20 employees and their respective salaries) and completed a manipulation check. After seeing all four offices, they indicated how much the company's overall distribution of wages was unequal, how fair it was, how concerned they were by it, and the extent to which inequality was a problem that merits organizational attention.

First, we examined whether the manipulation of segregation was successful. As intended, a two-tailed independent-sample t-test found that participants perceived less inequality within each office in the Economic Segregation condition ($M = 2.79$, $SD = 1.17$) than the Economic Integration condition ($M = 4.65$, $SD = 1.39$; two-tailed independent-sample $t(538.69) = 17.33$, $p < 0.001$, $d = 1.455$, 95% CI[1.65, 2.07]).

Next, we examined participants' judgments of the company's overall level of inequality. Replicating Study 2a, a series of two-tailed independent-sample t-tests found that the distribution of wages in the company was seen as less unequal, more fair, and less concerning in the Economic Segregation condition than the Economic Integration condition (Inequality: $M_{diff} = 0.91$, two-tailed independent-sample $t(575.52) = 6.47$, $p < 0.001$, $d = 0.522$, 95% CI[0.64, 1.19]; Fairness: $M_{diff} = -0.64$, $t(582) = -4.28$, $p < 0.001$, $d = 0.354$, 95% CI[−0.93, −0.34]; Concern: $M_{diff} = 0.63$, two-tailed independent-sample $t(582) = 3.95$, $p < 0.001$, $d = 0.336$, 95% CI[0.32, 0.95]). Participants also believed that wage inequality requires less attention in the Economic Segregation condition than the Economic Integration condition (Inequality is a big issue: $M_{diff} = 0.85$, two-tailed independent-sample $t(581.88) = 5.46$, $p < 0.001$, $d = 0.448$, 95% CI[0.54, 1.15]; Need to decrease inequality: $M_{diff} = 0.49$, two-tailed independent-sample $t(581.88) = 3.26$, $p < 0.001$, $d = 0.273$, 95% CI[0.20, 0.79]). Thus, despite seeing the same level of inequality, participants were less concerned when it mainly occurred between offices (i.e., economic segregation) than within offices.

Finally, we did not find any evidence of moderation by level of inequality. As shown in Table 3, a series of multiple regression analyses predicting judgments of inequality from condition (Economic Integration vs. Economic Segregation), level of inequality (Low, Medium, High, and Very High), and their interaction consistently found two main effects but no evidence of an interaction. Across different levels of inequality, participants were much less concerned about it when higher- and lower-paid employees worked in different offices.

## Study 3

In both between-participants and within-participant research designs, we consistently found that economic segregation reduces concern about inequality. Study 3 extends these findings to a hypothetical scenario related to a real-life context, examining how thinking about economic segregation affects lay people's concerns about inequality in one of the world's largest multinational technology companies.

Two hundred participants read about various positions at Amazon.com, including the estimated salary for each position. Specifically, participants saw information about 25 different positions at Amazon.com (e.g., Senior Product Manager, Cloud Support Associate, Warehouse packer, etc.), including each position's average annual salary as reported by Glassdoor—an online community website where employees anonymously share information about their employers, including their wages (this information is, of course, unofficial and therefore unverified). Participants were told that they would view five random jobs from each work location at a time and were randomly assigned to one of two conditions. In the Economic Segregation condition, the positions were segregated by pay, such that the five highest paid positions were separated from the second highest five positions, which were separated from the next highest 5 positions, and so forth. In the Economic Integration condition, the same positions were presented in a way that emphasized the contrast between them, with each location including higher and lower paid positions. Importantly, since

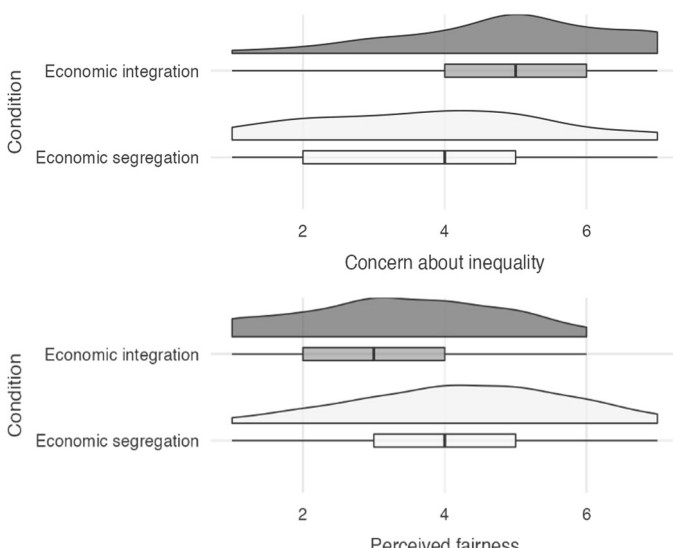

**Fig. 2 | Differences in concern about (top panel), and perceived fairness of (bottom panel), wage disparities between different job positions at Amazon.com when inequality was presented as mainly occurring between locations (Economic Segregation condition) or concentrated within locations (Economic Integration condition).** Box plots reflect 25th, 50th, and 75th percentiles; $N = 200$. (Study 3).

all participants saw the same 25 positions, both conditions had the same level of overall inequality and only varied in whether it occurred between (Economic Segregation) or within (Economic Integration) locations.

Participants saw five positions at a time (including each position's salary) and indicated, as a manipulation check, the level of inequality between positions. After seeing all 25 positions, participants indicated their concern about the overall level of inequality at Amazon.com and how fair it seemed to be. Finally, participants reported their familiarity with each of the 25 positions.

As before, the manipulation check revealed that participants saw less inequality within each quintet of positions in the Economic Segregation condition ($M = 3.21$, $SD = 0.88$) than the Economic Integration condition ($M = 4.99$, $SD = 1.09$; two-tailed independent-sample $t(188) = 12.34$, $p < 0.001$, $d = 1.791$, 95% CI[1.49, 2.96]).

Next, we examined how economic segregation impacted concerns about the overall level of organizational inequality. Two independent-sample t-tests revealed that participants were less concerned and saw Amazon.com as more fair when the inequality in wages mainly occurred between locations (Economic Segregation) than within each location (Economic Integration) (Concern: $M_{diff} = 1.18$, two-tailed independent-sample $t(188) = 5.16$, $p < 0.001$, $d = 0.748$, 95% CI [0.73, 1.63]; Fairness: $M_{diff} = -0.92$, two-tailed independent-sample $t(188) = -4.41$, $p < 0.001$, $d = 0.639$, 95% CI[-1.33, -0.51],) (Fig. 2). Moreover, we did not find any evidence of moderation by familiarity with the different job positions for either fairness perceptions of ($b = 0.046$, 95% CI[-0.011, 0.104], $t(186) < 1.59$, $p = 0.110$, partial $\eta^2 = 0.013$) or concerns about ($b = -0.044$, 95% CI[-0.107, 0.020], $t(186) = 1.356$, $p = 0.177$, partial $\eta^2 = 0.010$) economic inequality. Thus, even when considering positions at a real, multinational corporation, participants who saw inequality as mainly occurring between locations were much less concerned by it than those who saw it as occurring within locations.

## Study 4
We next explore whether actual levels of economic segregation affect concerns about inequality beyond the realm of organizational pay disparity. Specifically, using a real-life context, we examine the

generalizability of the effect of economic segregation in judgments about a country with a high level of economic inequality—South Africa.

Five hundred three participants saw aerial photos of South African cities (Johannesburg, Cape Town, Pietermaritzburg, and Durban) and were randomly assigned to view four higher-income and four lower-income neighborhoods either side-by-side or separated from each other. In the Economic Integration condition, participants saw photos of higher-income and lower-income neighborhoods that visually emphasized the juxtaposition between higher-income and lower-income South Africans. In the Economic Segregation condition, they saw the same photos, edited to appear as if they were separated based on income (i.e., higher-income neighborhoods appeared in separate screens from lower-income neighborhoods). On each screen, participants completed a manipulation check indicating the level of inequality in the photo. Then, after seeing all eight neighborhoods, participants reported their views about the overall level of inequality in South Africa: how concerned they were by the distribution of resources in the country, how fair they perceived it to be, and how much they were willing to tolerate it. Thus, both conditions presented the same eight neighborhoods and only differed in whether the pictures were edited to show higher- and lower-income neighborhoods as separated from (Economic Segregation) or adjacent to (Economic Integration) each other.

First, we examined whether the manipulation affected perceptions of inequality. As intended, a two-tailed independent-sample t-test found that participants viewed much less inequality in the Economic Segregation condition ($M = 3.69$, $SD = 0.91$) than the Economic Integration condition ($M = 5.50$, $SD = 1.27$; two-tailed independent-sample $t(435.24) = 17.78$, $p < 0.001$, $d = 1.626$, 95% CI[1.60, 2.01]).

We next examined whether visually separating higher- and lower-income neighborhoods reduced concerns about inequality. Indeed, a series of two-tailed independent-sample t-tests revealed that participants worried less about inequality, perceived it as more fair, and were more tolerant of it in the Economic Segregation condition than the Economic Integration condition (Concern: $M_{diff} = 0.96$, $t(470) = 6.27$, $p < 0.001$, $d = 0.577$, 95% CI[0.66, 1.26]; Fairness: $M_{diff} = -0.55$, $t(458.91) = -3.77$, $p < 0.001$, $d = 0.346$, 95% CI[-0.84, -0.26]; Tolerance: $M_{diff} = 0.51$ $t(470) = 4.90$, $p < 0.001$, $d = 0.451$, 95% CI[0.30, 0.71]; all two-tailed independent-sample tests). Since all participants saw the same neighborhoods, this result cannot be explained by different exposure to inequality. Rather, segregation in and of itself reduced concerns about inequality. That is, seeing higher- and lower-income neighborhoods as separated from each other reduced concerns about otherwise identical levels of inequality.

## Study 5a
Both real and hypothetical levels of economic segregation in wages and neighborhoods reduced concerns about otherwise identical levels of inequality. Study 5a replicates these findings with a new design and examines one proposed mechanism for this effect. Beyond distorting perceptions of inequality, we examine whether a hypothetical level of high economic segregation reduces concerns about inequality through an additional pathway: reduced social comparisons.

Participants (N = 303) imagined living in each of four cities of a given country. For each city, participants saw images of a busy street and the incomes of 12 individuals who live in that city, amounting to a total of 48 individuals across all four images. We randomly assigned participants to one of two conditions, keeping the overall level of inequality across cities constant but varying whether it mostly occurred within each city or between them. In the Economic Segregation condition, participants saw a segregated country where inequality mainly occurred between cities (i.e., the highest-income people live in a different city from second highest-income people, who live in a different city from the next highest-income people, and so on). In the Economic Integration condition, participants saw a country where

inequality mainly occurred within each city (i.e., the highest and lowest income residents live in the same cities). Thus, participants saw the same level of overall inequality (as measured by the dispersion of incomes among all 48 people) but varied in whether it mostly occurred between cities (Economic Segregation condition) or within cities (Economic Integration condition).

As a manipulation check, participants rated the level of inequality within each city. Then, after seeing all four cities, participants considered the country as a whole, rating the fairness of the distribution of income in the country and their concerns about it. Finally, to examine the role of economic comparisons, participants imagined living in this country and indicated how often they would think about the amount of money that they and other people make.

As before, the manipulation was successful, leading participants to see lower inequality in the Economic Segregation condition ($M = 2.56$, $SD = 1.34$) than the Economic Integration condition ($M = 4.78$, $SD = 1.25$; two-tailed independent-sample $t(286) = 14.54$, $p < 0.001$, $d = 1.714$, 95% CI[1.92, 2.52],).

Next, we examined whether segregation affected concerns about inequality. As predicted, two-tailed independent-sample t-tests found that participants were less worried about inequality when high- and low-income people lived in different cities (Concern: $M_{diff} = 1.12$, two-tailed independent-sample $t(282.81) = 5.03$, $p < 0.001$, $d = 0.591$, 95% CI[0.68, 1.55]; Fairness: $M_{diff} = -0.67$, two-tailed independent-sample $t(286) = -3.14$, $p = 0.002$, $d = 0.370$, 95% CI[−1.10, −0.25]). Despite viewing the same level of inequality, participants were much less concerned when it mainly occurred between higher- and lower-income cities rather than within each city.

Finally, we examined whether economic comparisons mediated the effect of segregation on concerns about inequality. As predicted, participants expected to think less about their and others' income in the Economic Segregation condition than the Economic Integration condition ($M_{diff} = 0.54$, two-tailed independent-sample $t(279.34) = 3.26$, $p = 0.001$, $d = 0.383$, 95% CI[0.21, 0.87]), which partially mediated the effect of segregation (Concern indirect effect: $\beta = 0.49$, $SE = 0.15$; 95% CI[0.20, 0.79], direct effect: $\beta = 0.63$, $SE = 0.17$; 95% CI[0.30, 0.96]; Fairness indirect effect: $\beta = -0.28$, $SE = 0.09$; 95% CI[−0.46, −0.10], direct effect: $\beta = -.40$, $SE = 0.20$; 95% CI[−0.79, 0.00]). Thus, participants expected to compare their and others' income less frequently in economically segregated cities and, as a result, were less concerned by the level of inequality in the economically segregated country.

## Study 5b

We examined the robustness of this effect in a direct replication with a new sample of 301 participants. As in Study 5a, participants were less worried about inequality in the Economic Segregation condition (when financially better-off and worse-off people lived in different cities) than the Economic Integration condition (Concern: $M_{diff} = 0.55$, two-tailed independent-sample $t(287.83) = 2.49$, $p = 0.013$, $d = 0.292$, 95% CI[0.12, 099]; Fairness: $M_{diff} = -0.54$, two-tailed independent-sample $t(288) = -2.58$, $p = 0.010$, $d = 0.303$, 95% CI[−0.95, −0.13]). In addition, participants expected to think less about their and others' income in the Economic Segregation condition than the Economic Integration condition ($M_{diff} = 0.70$, two-tailed independent-sample $t(288) = 4.40$, $p < 0.001$, $d = 0.516$, 95% CI[0.38, 1.01]), and this difference in expected comparisons mediated the effect on participants' concern about inequality (indirect effect: $\beta = 0.41$, $SE = 0.11$; 95% CI[0.20, 0.61], direct effect: $\beta = 0.15$, $SE = 0.21$; 95% CI[−0.26, 0.56]). Finally, as an alternative account, we examined whether economic segregation assuages concerns about inequality by fostering internal attributions of economic circumstances[5]. In contrast to this alternative account, participants were less (not more) prone to make internal attributions in the Economic Segregation condition than the Economic Integration condition ($M_{diff} = -8.62$, two-tailed independent-sample

$t(287) = 2.82$, $p = 0.005$, $d = 0.332$, 95% CI[−14.63, −2.60]), suggesting that segregation reduces concerns about inequality by dampening economic comparisons, not by heightening internal attributions of economic outcomes.

## Discussion

Across a series archival and experimental studies with online and lab-based research samples and using between- and within-participants research designs, we found that both real and hypothetical economic segregation reduce concerns about inequality. Even when people were exposed to identical levels of inequality, segregation among a country's residents or a company's employees weakened concerns about it. Thus, by obscuring the contrast between people of different means, economic segregation can lead people to be less concerned about economic inequality than they would otherwise.

It is important to emphasize that economic segregation reduces concerns about inequality even when people see the same levels of it. Although misperceptions of inequality abound[8,45,46], our findings suggest that economic segregation reduces concerns about inequality independent of such misperceptions. Complementing past research (which focused on what people know about inequality; 12–17, 23), we found an effect of segregation even when people were exposed and explicitly reported seeing equal levels of inequality (Studies 2b and 2c). Thus, by showing participants the same distribution of wages and merely manipulating whether the disparity was exhibited within each office (economic integration) or between offices (economic segregation), we were able to isolate the effect of segregation independent of any misperceptions of inequality. Consequently, economic segregation likely impacts attitudes about inequality through two distinct pathways: by shaping perceptions of how much inequality exists in society and by impeding economic comparisons. While future research should examine the boundary conditions of these pathways, our findings make one thing clear: economic segregation reduces concerns about inequality above and beyond people's perceptions of its magnitude.

Of course, other factors beyond economic segregation may similarly weaken concerns about inequality by reducing the juxtaposition of wealth and poverty. Indeed, since the juxtaposition of wealth and poverty is most likely a multi-faceted phenomenon, future research could benefit from examining the various factors that influence when and why people are impeded from engaging in economic comparisons, including a lack of exposure to news articles about economically varied regions, working with an economically homogenous customer base, attending private (vs. public) institutions of higher education, using private (vs. public) means of transportation, and so forth. Thus, future research could explore additional factors that reduce the juxtaposition of wealth and poverty and, as a result, concerns about inequality.

Although we focused on economic segregation, our findings may shed light on other forms of segregation. Despite abolishing formal racial segregation, informal segregation persists both in the U.S. and in post-Apartheid South Africa[47,48], among other places. Similarly, an analysis of public spaces in Northern Ireland revealed that, despite the lack of formal barriers, Catholics and Protestants often limit their use of outgroup-centric spaces[49]. Thus, since socioeconomic status often intersects with people's racial and ethnic identities, the effect of economic segregation on concern about inequality may be amplified by group-based segregation. If so, this may help explain why wealthy White Americans are especially prone to underestimate racial disparities[50], which further facilitates their perpetuation.

Despite documenting a consistent pattern, each study may have its specific limitations. Since people work, socialize, and send their children to school near their home[19,20,23,42], Study 1 relied on available, large-scale, correlational data of residential economic segregation. While we addressed this issue in Studies 2a-2d (where the effect of

segregation was found in non-residential domains), future research could examine how other types of segregation affect attitudes about inequality. Second, while the archival nature of Study 1 merits the usual caveats regarding correlational data, this concern is alleviated by the experimental design of Studies 2–5 and the effect's robustness across virtually all model specifications. Nevertheless, future work could examine whether people who care less about inequality opt to live in more segregated areas, creating a bidirectional relationship between segregation and concerns about inequality. Third, despite finding consistent evidence in hypothetical and real-world scenarios, economic segregation 'in the wild' may interact with other factors that were not captured by the current methods and so caution is warranted when drawing conclusions to large-scale social concerns. At the same time, Study 1 complements these experimental findings by showing through the use of archival data the real-world correlation between economic segregation and concerns about inequality. Finally, while we focused on the effect of economic segregation within the U.S. (due to a lack of cross-country data), future research could examine its effect in other countries and cultural contexts.

Notably, we did not find statistically significant evidence of moderation by income, and it remains to be seen how segregation affects people across the income ladder. Since segregation reduces the visibility of others for both financially well-off and financially worse-off people, it likely impacts concerns about inequality regardless of one's own economic standing. At the same time, the effects of segregation may not necessarily translate to similar actions across the income ladder (e.g., in people's willingness to sign anti-inequality petitions; 15,16). And, while economic integration may increase wealthy people's concern about inequality, their views of the world as meritocratic[5,6] and of wealth as non-zero-sum[39,51] may still dampen their support of inequality-reducing policies[52]. Thus, to understand such nuances, future research should examine the effects of economic segregation on people across the socioeconomic ladder.

Finally, although we focused on one specific pathway through which segregation reduces concerns about inequality (i.e., economic comparisons), future research could examine the relationship between different pathways. Clearly, economic segregation shapes attitudes about inequality through two independent and distinct pathways: by distorting perceptions of inequality and by reducing the comparisons people make between individuals of different means. Yet, the interplay between these pathways remains to be explored. For instance, by underestimating the level of inequality in society, people may be less prone to compare individuals of different means to each other, and thus be less concerned about it. Moreover, such obstacles to comparing people of different means may further distort perceptions of inequality, leading people to underestimate it. Thus, by examining the reciprocal relationship between these two pathways, future research could further our understanding of why economic segregation reduces concerns about inequality.

The growing interest in how people perceive, make sense, and react to inequality has mainly focused on reactions to economic disparities but has failed to account for the physical and functional distributions of such disparities. Yet inequality often hides in plain sight and can be obscured by the fault lines of segregation. As long as people of different economic means live in different areas, work at different companies, send their children to different schools, and engage in different leisure activities, they may be less likely to care about inequality.

## Methods

For all studies, sample sizes were determined in advance, analyses were conducted after data collection was complete, and we report all measures and conditions. The research project was approved by the IRB committee at Columbia University. Participants in all of the experimental studies completed an informed consent prior to commencing their participation.

We pre-registered five studies:
- Study 2A (10/22/2021) - https://aspredicted.org/3zw2d.pdf
- Study 2B (07/25/2023) - https://aspredicted.org/cc5gm.pdf
- Study 2D (11/12/2021) - https://aspredicted.org/rc557.pdf
- Study 3 (08/14/2023) - https://aspredicted.org/ap8e9.pdf
- Study 5A (12/21/2021) - https://aspredicted.org/3yc6k.pdf

### Study 1

**The data.** We combined data from two unique archival datasets to examine whether segregation predicts concern about inequality. The economic segregation data was aggregated from sample data of the 2000 U.S. Census across 741 Commuter Zones (i.e., clusters of counties with strong commuting ties; 43).

**The measures.** The segregation data includes three measures of economic segregation for each Commuter Zone: (1) the overall degree to which households above a given percentile of the local income distribution are geographically segregated from households below that percentile, (2) the degree to which high-income households (>25th percentile) are segregated from economically worse-off households, and (3) the degree to which low-income households (<75th percentile) are segregated from better-off households. In addition, this dataset includes three measures of economic inequality for each Commuter Zone: (1) the overall Gini coefficient, (2) the Gini coefficient of the bottom 99% of households in the area, and (3) the income shares of the Top 1% of households in the area. Finally, the dataset includes measures of population size and income per capita. All variables were standardized across the entire dataset.

In the World Value Survey data, we focused on responses to a single-item measure of attitudes about inequality, reverse-scored such that higher values correspond to higher concern about inequality (on a 10-point scale, whether respondents believe that 'Incomes should be made more equal' versus 'We need larger income differences as incentives').

**The analyses.** As described in the main text, we examined the robustness of the effect of economic segregation on concern about inequality by estimating all potential models using all possible combinations of the three segregation measures and three inequality measures, both when including and when omitting additional control variables (state population, income per capita, proportion of Black residents, racial segregation, and proportion of votes for the Democratic party in the 2020 elections). As a further robustness check, we examined all 18 different combinations of these parameters (3 measures of segregation x 3 measures of inequality x 2 options of whether or not to include control variables) by excluding data from each state at a time, producing a total of 918 different estimates (18 different combinations x 51 exclusions, including a model with no exclusions). For further details, see the Supplemental Materials.

### Study 2a

**Participants.** Two hundred ninety-nine U.S. residents were recruited from Amazon's Mechanical Turk to participate in a pre-registered study. We excluded 9 participants who failed an attention check, leaving a sample of 290 participants (143 female, 144 male, 2 non-binary, 1 Other; $M_{age}$ = 42.72; 80.3% White, 6.4% Asian, 8% Black/African American, 3% Latinx/Hispanic, 0.7% American Indian/Alaska Native, 0.3% Native Hawaiian/Pacific Islander, 1.3% Other). A power analysis revealed 80% power to detect an effect size of d = 0.33 or larger in a between-participants design.

**Procedure.** Participants read about a company in a large metropolitan area, saw pictures of 80 employees and their annual salaries (across

four offices), and were randomly assigned to one of two conditions. In the Economic Integration condition, the organizational inequality mainly occurred within each office, such that every office employed some of the highest and lowest-paid employees. In the Economic Segregation condition, the four offices were relatively equal in terms of pay, with inequality mainly occurring between (rather than within) offices (i.e., 20 highest-paid employees were employed in one office, the second 20 highest-paid employees were employed in a second office, the next 20 employees were employed in a third office, and the 20 lowest paid employees were employed in a fourth office). For each office, participants completed a manipulation check (To what extent would you think that the distribution of wages in this office is equal or unequal? 1-Extremely equal, 7-Extremely unequal).

**Measures.** After seeing all wages across the four offices, participants indicated their views about the overall level of inequality in the company: the extent to which they perceived the organizational distribution of wages as unequal (To what extent would you think that the distribution of wages in this company is equal or unequal? 1-Extremely equal, 7-Extremely unequal), how fair they perceived it to be (How fair or unfair would you view the distribution of wages in this company to be? 1-Extremely unfair, 7-Extremely fair), how concerned they were by it (How concerned would you be about the distribution of wages in this company? 1-Not concerned at all, 7-Extremely concerned), and how much they saw it as a problem that merits organizational attention (I believe that wage inequality is a big issue for workers in this company, I believe that this company should work hard to implement policies that decrease the wage disparities between employees; 1-Strongly disagree, 7-Strongly agree).

### Study 2b
**Participants.** One hundred U.S. residents were recruited from Prolific Academic to participate in a pre-registered study. We excluded 8 participants who failed an attention check, leaving a sample of 92 participants (44 female, 45 male, 2 nonbinary, 1 Other; $M_{age} = 41.91$; 77.8% White, 4.6% Asian, 8.4% Black/African American, 5.6% Latinx/Hispanic, 1.9% American Indian/Alaska Native, 1.9% Other). A power analysis revealed 80% power to detect an effect size of dz = 0.30 or larger in a within-participants design.

**Procedure.** Participants saw pictures of 20 employees of a company (and their annual salaries). In counterbalanced order, they saw these employees grouped across four offices in two different ways: an economically integrated manner (i.e., organizational inequality mainly occurred within each office, such that every office employed some of the highest and lowest paid employees) and an economic segregated manner (i.e., organizational inequality mainly occurred between offices, such that the five highest paid employees were employed in one office, the next five highest paid employees were employed in a second office, the next five employees were employed in a third office, and so on). For each office, participants completed a manipulation check (To what extent is the distribution of wages in each of the following offices equal or unequal? 1-Extremely equal, 7-Extremely unequal).

**Measures.** Participants indicated their level of concern about the overall level of inequality in the company (How concerned would you be about the distribution of wages in this company? 1-Not concerned at all, 7-Extremely concerned) and how fair they perceived it to be (How fair or unfair would you view the distribution of wages in this company to be? 1-Extremely unfair, 7-Extremely fair). Following, they were asked to look at all of the wages in company across all four offices and to indicate their perception of the overall level of inequality in the organization (To what extent is the distribution of wages in this entire company equal or unequal? 1-Extremely equal, 7-Extremely unequal). Finally, after completing these measures for one method of grouping

(i.e., economically segregated/integrated), participants completed the same procedure for the alternative method of grouping.

### Study 2c
**Participants.** One hundred seventeen participants were recruited by a research lab at Duke University. We excluded six participants who failed an attention check, leaving a sample of 111 participants (84 female, 26 male, 1 nonbinary, 1 Other; $M_{age} = 35.28$; 56.6% White, 26.2% Asian, 9% Black/African American, 4.9% Latinx/Hispanic, 1.6% Native Hawaiian/Pacific Islander, 1.6% Other). A power analysis revealed 80% power to detect an effect size of dz = 0.27 or larger in a within-participants design.

**Procedure and measures.** The study procedure, materials, and measures were identical to Study 2b.

### Study 2d
**Participants.** Six hundred two U.S. residents were recruited from Amazon's Mechanical Turk to participate in a pre-registered study. We excluded 18 participants who failed an attention check, leaving a sample of 584 participants (318 female, 256 male, 6 nonbinary, 4 Prefer not to say; $M_{age} = 42.80$; 75.3% White, 8.9% Asian, 7.4% Black/African American, 5.5% Latinx/Hispanic, 0.9% American Indian/Alaska Native, 0.8% Native Hawaiian/Pacific Islander, 1.2% Other). A power analysis revealed 80% power to detect an effect size of d = 0.23 or larger in a between-participants design.

**Procedure.** The study was identical to Study 2a, except for the added manipulation of level of organizational inequality. Participants were randomly assigned to one of eight conditions in a 2 (Condition: Economic Integration vs. Economic Segregation) × 4 (Inequality: Low, Medium, High, and Very High) between-participants design. As before, they saw 80 employees' salaries across four offices in one of two conditions varying in whether wage disparity mainly occurred within each office (Economic Integration condition) or between different offices (Economic Segregation condition). Participants were further assigned to one of four inequality conditions, varying in the overall level of organizational inequality [Low (Gini = 0.22), Medium (Gini = 0.32), High (Gini = 0.42), or Very High (Gini = 0.52)]. Importantly, within each level of inequality, the distribution of wages was identical in both conditions and only differed in whether it occurred within or between offices. For each office, participants completed a manipulation check (To what extent would you think that the distribution of wages in this office is equal or unequal? 1-Extremely equal, 7-Extremely unequal).

**Measures.** After seeing all the wages across the four offices, participants indicated their views of the overall level of inequality in the organization using the same items from Study 2a.

### Study 3
**Participants.** Two hundred U.S. residents were recruited from Prolific Academic to participate in a pre-registered study. We excluded 10 participants who failed an attention check, leaving a sample of 190 participants (89 female, 91 male, 6 nonbinary, 4 Other; $M_{age} = 38.36$; 77.5% White, 68% Asian, 6% Black/African American, 5.5% Latinx/Hispanic, 0.5% American Indian/Alaska Native, 0.5% Native Hawaiian/Pacific Islander, 2% Other). A power analysis revealed 80% power to detect an effect size of d = 0.41 or larger in a between-participants design.

**Procedure.** Participants learned about 25 job positions at Amazon.com, including the average annual salary for each position, as reported by Glassdoor—an online community website where employees anonymously share information about their employers, including

their wages (this information is unofficial and has not been independently verified). They were randomly assigned to one of two conditions, in which they saw five different positions at a time. In the Economic Integration condition, the inequality between the different positions was presented as mainly occurring within each work location, such that each location included some of the highest and lowest paid positions. In the Economic Segregation condition, the work locations were presented as relatively equal in terms of pay, with inequality across positions mainly occurring between (rather than within) locations (i.e., the five highest paid positions were presented separately from the second five highest paid position, which were presented separately from the next five highly paid position, and so forth). After seeing each five positions, participants completed a manipulation check (To what extent would you think that the distribution of wages between the different positions is equal or unequal? 1-Extremely equal, 7-Extremely unequal).

**Measures.** After seeing all 25 job positions, participants indicated their views about the overall level of inequality in the company: how concerned they were by the inequality in wages across positions it (How concerned are you about the distribution of wages in this company? 1-Not concerned at all, 7-Extremely concerned) and how fair they perceived it to be (How fair or unfair is the distribution of wages in this company? 1-Extremely unfair, 7-Extremely fair). Finally, participants saw a list of all 25 positions, and indicated whether they were familiar with each one (I am/am not familiar with this position).

## Study 4
**Participants.** Five hundred three U.S. residents were recruited from Amazon's Mechanical Turk. We excluded 31 participants who failed an attention check, leaving a sample of 472 participants (249 female, 219 male, 2 Other; $M_{age}$ = 43.52; 76.3% White, 5.2% East Asian, 1.2% Southeast Asian, 8.6% Black/African American, 4.6% Latinx/Hispanic, 1.9% American Indian/Alaska Native, <1% Middle-Eastern/Arab, 1.2% other). A power analysis revealed 80% power to detect an effect size of d = 0.26 or larger in a between-participants design.

**Procedure.** Participants saw aerial images of four South African cities (Johannesburg, Cape Town, Durban, and Pietermaritzburg; retrieved from http://unequalscenes.com) and were randomly assigned to one of two conditions. In the Economic Integration condition, participants saw four pictures of adjacent high- and low-income neighborhoods, emphasizing the actual physical proximity of higher- and lower-income South Africans. In the Economic Segregation condition, they saw the same eight neighborhoods, edited to reduce the salience of the actual proximity of higher- and lower-income South Africans (i.e., high- and low-income neighborhoods were presented on separate screens). For each pair of neighborhoods, participants completed a manipulation check (To what extent do you think that the distribution of economic resources in these neighborhoods is equal or unequal? 1-Extremely equal, 7-Extremely unequal).

**Measures.** After viewing all eight neighborhoods, participants indicated their views about the overall level of inequality in South Africa: how fair it seemed to be (In your opinion, how fair or unfair is the distribution of economic resources in South Africa? 1-Extremely unfair, 7-Extremely fair), how concerned they were by it (How concerned are you about the distribution of economic resources in South Africa? 1-Not concerned at all, 7-Extremely concerned), and how much they were willing to tolerate it (using the five-item Support for Economic Inequality Scale; e.g., We need to do everything possible to reduce economic inequality in South Africa today; α = 0.86;[53]). Finally, participants completed an exploratory measure of charitable donations and reported their age, gender, ethnicity, household income, and ideology.

## Study 5a
**Participants.** Three hundred three U.S. residents were recruited from Amazon's Mechanical Turk to participate in a pre-registered study. We excluded 15 participants who failed an attention check, leaving a sample of 288 participants (167 female, 120 male, 1 Other; $M_{age}$ = 41.20; 70.4% White, 6% East Asian, 2.8% Southeast Asian, 11.7% Black/African American, 5.1% Latinx/Hispanic, 1.9% American Indian/Alaska Native, 1.3% Middle Eastern/Arab, 1% Other). A power analysis revealed 80% power to detect an effect size of d = 0.33 or larger in a between-participants design.

**Procedure.** Participants saw four images of busy cityscapes in an unnamed country. In each picture, they saw the annual salaries of 12 people and were randomly assigned to one of two conditions. In the Economic Integration condition, the inequality within each city was relatively high (i.e., both high- and low-income residents lived in the city). In the Economic Segregation condition, the inequality was mainly distributed between cities, such that one city had only very high-income residents, another city had only average-to-high income residents, a third city had only average-income residents, and a fourth city had only low-income residents. For each city, participants completed a manipulation check (To what extent do you think that the distribution of wages in this city is equal or unequal? 1-Extremely equal, 7-Extremely unequal).

**Measures.** After seeing all four cities, participants indicated their views about the country's overall level of inequality: how fair it was (How fair or unfair would you view the distribution of wages in this country to be? 1-Extremely unfair, 7-Extremely fair) and how concerned they were by it (How concerned would you be about the distribution of wages in this country? 1-Not concerned at all, 7-Extremely concerned). Finally, they reported how often they would engage in economic comparisons if they lived in this country, indicating their tendency to think about their own income (If you were one of the people living in this country, how often would you think about how much money you make?) and their tendency to think about other people's income (If you were one of the people living in this country, how often would you think about how much money other people make?) (1-Not often at all, 7-Very often; r = 0.64).

## Study 5b
**Participants.** Three hundred one U.S. residents were recruited from Amazon's Mechanical Turk to participate. We excluded 11 participants who failed an attention check, leaving a sample of 290 participants (119 female, 169 male, 2 Other; $M_{age}$ = 43.46; 75.4% White, 6.1% East Asian, 1.9% Southeast Asian, 11.5% Black/African American, 2.6% Latinx/Hispanic, 1.6% American Indian/Alaska Native, <1% Other). A power analysis revealed 80% power to detect an effect size of d = 0.33 or larger in a between-participants design.

**Procedure.** Study 5b followed the exact same procedure as Study 5a with one important addition: After completing a measure of economic comparison, participants also completed a measure of economic attribution, indicating on a 100-point sliding scale the extent to which they believe that in the country they just saw people's economic circumstances are due to 'external circumstances beyond their control- 0' versus 'their own efforts – 100.'

### Reporting summary
Further information on research design is available in the Nature Portfolio Reporting Summary linked to this article.

## Data availability
The materials and datasets generated for Studies 2a-5b have been deposited in the Open Science Framework (OSF) repository and are accessible here: 10.17605/OSF.IO/U2T7D. Study 1 used two publicly available datasets: the World Values Survey (www.worldvaluessurvey.

org) and the Opportunity Insights dataset (including data from the 2000 U.S. census; https://rajchetty.com/wp-content/uploads/2021/04/land-of-opportunity.html-charsetutf-8).

## Code availability

The code for the analyses of Study 1 is available online through the Open Science Framework (https://doi.org/10.17605/OSF.IO/U2T7D).

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

## Acknowledgements

The authors have no external funding to report.

## Author contributions
S.D., D.G.T., and M.A. have contributed equally to the manuscript.

## Competing interests
The authors declare no competing interests.
