## [Peer Review File · Nature Communications]

Economic segregation reduces concern about economic inequalityReviewers' Comments:

Reviewer #1:

Remarks to the Author:

The article "Economic segregation reduces concern about economic inequality" analyses the perception of inequality along different dimensions mainly through experimental methods by varying how different neighbourhoods/people are presented together or separately. I think the authors do a great job of conducting a set of experiments that try to get at the same pattern from different angles. However, there are two concerns, one of which can be alleviated easily by reformulating some parts of the study, while the other might be more difficult for the authors and involve additional data collection.

1. Theory/interpretation: It seems that the authors come from a very experiment-focussed background (my apologies if this is not correct). I, on the other hand, tend to use mainly survey data, which might explain this critique:

I think the experiments show interesting patterns of participant's responses being exposed to different sets of numbers or images. However, I think this is what is primarily happening, people react to numbers and pictures. This is, per se, not bad, but I would caution a more careful interpretation of the findings. Connecting these very abstract experimental settings to large-scale social concerns about preferences for policies about inequality and residential segregation jumps at least one step. I would find the argument more convincing that starts by the likely real-world associations that participants have when completing the experiments and extrapolating what this can mean for their views about the world. E.g. what company do people think about that has extreme inequality within offices, compared to one that has inequality between offices? What real-world associations might these examples elicit and what can be learned from this?

I think this study has some interesting insights, especially given that different experiments were combined, but I would be more convinced by an interpretation from the bottom and a more careful conclusion about what this means for country-wide concerns about inequality.

2. Methodology: The study was conducted using MTurk. This, unfortunately, means that the authors have little control over data quality. Recent reports (e.g. <https://journals.sagepub.com/doi/10.1177/17456916221120027>) about the rate of valid responses that actually engage with an experiment in a meaningful way are disheartening. It is, therefore, possible that some or many of the patterns were produced by participants that only had passing attention to the questions and reacted, e.g., on the most recent picture or condition. This would question the validity of the findings.

The preregistration document stipulates that "Participants will be excluded for failing simple attention check questions", which is a good start. However, regular MTurk participants should be able to spot them by now and be able to answer them properly even when trying to minimize the time spent. Having some replication of the results with an in-person pool of experimental participants in a physical lab for whom ensuring data quality is easier would greatly alleviate my concerns that the results reflect people engaging with questions, rather than clicking through a survey as quick as possible to make money.

Reviewer #2:

Remarks to the Author:

The article builds an interesting narrative connecting economic segregation and the perception of inequality. Its main objective is to test the hypothesis that individuals from segregated economies are less concerned about economic inequality. To achieve this, the authors employed a combination of one quantitative analysis and five qualitative experiments, using a hypothetical company, city, satellite images from a highly unequal country, and archival datasets. Based on their findings, the authors conclude that individuals from less segregated environments exhibit greater concern about inequality than their more segregated counterparts. They propose that the juxtaposition of inequality in less

segregated scenarios, which disappears in segregated scenarios, is the main force driving a higher concern about inequality.

Overall, this is a strong paper potentially suitable for Nature Communications-- however, I would love to see implemented the issues discussed below before a final assessment is made.

General assessment of the work

The authors should provide additional justification for their claim that economic segregation eliminates the juxtaposition of inequality for both poor and rich individuals. A notable concern is the lack of references to support the argument that economically-segregated societies display less concern about inequality due to the inhibiting effect of segregation on the comparison between the rich and the poor in various aspects such as work, education, and housing. The central hypothesis and conclusion of the manuscript rely on this assumption, which has yet to be adequately supported in the introduction and discussion sections. For instance, the example provided (lines 75-77) of an expensive meal and a homeless person is used to support the hypothesis that the rich and the poor are not juxtaposed in segregated societies. However, it can be argued that in highly unequal countries, middle-upper-class restaurants do have street vendors entering, thus challenging this example. Similarly, in the discussion (line 358), the same assumption is made without being substantiated with references.

While the authors effectively demonstrate that the absence of juxtaposition between economic classes diminishes concerns about inequality, it remains unclear whether economic segregation itself is responsible for this reduction. The study convincingly supports the notion that when individuals from different economic backgrounds are physically separated and lack direct exposure to one another, their concerns about inequality may diminish. However, the specific causal relationship between economic segregation and complete lack of juxtaposition requires further clarification. The manuscript would benefit from additional evidence or theoretical reasoning to establish a clearer link between these two concepts.

Major comments

- A concern arises regarding Study 1, which utilizes data on economic segregation that is primarily based on residential segregation. This limited focus on residential segregation paints an incomplete picture of overall segregation encompassing various domains such as work, school, church, and others, which the authors intend to incorporate. To strengthen their argument, the authors should provide further justification for how residential segregation adequately represents economic segregation across society, as stated in the conclusion (lines 377-380).

Minor comments

- I 227,228: Authors write that participants believed that wage inequality requires more attention in the Economic Segregation condition than in the Economic Integration condition but it should be the other way around.

- I251 and I289: Manipulation check to rate inequality for each city/neighborhood shows less perceived inequality in the economic segregation scenario. Authors should comment on how this does not bias the overall perception of inequality in the country.

Reviewer #3:

Remarks to the Author:

The authors present one correlational archival study showing a robust negative association between local economic segregation and concern about economic inequality, four experiments using hypothetical vignettes to show people have greater concern over inequality in less economically

segregated companies and cities, and one additional experiment finding similar results using aerial photos of unequal neighborhoods in South Africa.

There is a lot to like about this paper. It addresses an important topic with an interesting hypothesis. Study 1's analysis of testing various sets of variable combinations is an excellent way to increase confidence and reduce researcher degrees of freedom in this study. And I especially liked the Mach band analogy.

At the same time, I feel the paper is significantly undermined by how the authors misrepresent, or perhaps simply misframe their findings. Over and over again throughout the manuscript, the authors emphasize that what differentiates this work from the prior literature is that their effects occur "holding constant what people know about inequality" and "even when people are fully aware of otherwise identical levels of economic disparities between the rich and the poor" allowing them "to isolate the direct effect of segregation on attitudes about inequality independent of any misperceptions." The central claim, as I understand it, is that prior work has shown that people are more bothered about inequality when they perceive more of it. But, even when people perceive the same amount of inequality, their concern over it will be directly affected by how well-integrated or segregated those inequalities are. My issue is that in every one of their experiments, people in the economically integrated condition perceive there to be more inequality than those in the economically segregated condition. Indeed, taking a quick peek at the Study 3 data (I appreciate the authors making all their data and materials available and easy to navigate), the difference in the perceived level of inequality is larger between the economically segregated and integrated conditions than it is between the Gini conditions, in which inequality actually differed. Moreover, the perception of inequality fully mediates the effect of the economic segregation manipulation on the concern about inequality, on fairness, and on support for action.

What prior work, like that of Sands, shows is that when the salience of inequality is raised (often via local instances of inequality), people perceive more inequality, and their concern affected. This work shows something similar, but does so by finding that by presenting people with different wages or levels of wealth in different local groupings, it changes how much global inequality is perceived, and in turn people's concern is affected.

Now this is an interesting finding in its own right, but it seems like it is an importantly different one from the one the authors present in the paper. To match the conclusions that the authors draw in the paper, I would think that the authors would want to show that the concern dependent measures are higher in economically integrated conditions even when participants recognize that the inequality is objectively the same. Take the lovely Mach bands example that the authors use. What makes that such a compelling demonstration is that we know that the bands are the same colors when presented together and apart, but fall for the illusion nonetheless. In these studies, however, people don't think that the inequality is the same.

If the authors wish to test the conclusion they espouse, they could construct a study in which, just like people would report explicitly understanding the Mach Band to be the same colors in the two presentations, people explicitly recognize the global inequality to be the same. For instance, they could first show the full group of wages, and then show participants how this same population can be grouped in different ways. One would assume that people would explicitly report the inequality be the same, but one could test whether different levels of concern emerge depending on how segregated or integrated the groupings ended up being.

Alternatively, the authors could change the framing of their paper to downplay the admittedly central claim that different levels of concerns emerge "even when people are fully aware of otherwise identical levels of economic disparities". Instead they could focus on the more cognitive process of how the different local grouping affects perceptions of global inequality. Although I'm not sure that would be as interesting as how the authors are currently framing things.

Minor points:

In the discussion, the authors write "Notably, since economic segregation reduces the visibility of "the other" for both the rich and the poor, its effects were not moderated by income." I might have missed the discussion of a lack of moderation by income. Was that for the first study? The subsequent studies seem less relevant since the income of the participant has nothing to do with the people or situation they are rating—which is the important question. Given Sands' findings that the rich and poor respond differently in response to exposure to inequality, it would be very interesting to see whether similar effects are seen here. That said, one would want to make the ratings relevant to the participants, so that they had "skin in the game" as rich and poor people do in real life.

For Study 4, it would be great to include examples of the stimuli so that readers could get a sense for themselves how differently economically integrated images feel compared to segregated ones.

Reviewer #4:

Remarks to the Author:

In this paper, the authors report the results of 6 studies (1 archival study and 4 experiments, with an additional direct replication) that aimed to explore the impact of economic segregation on people's perceptions of, and concern about, economic inequality. In line with their expectations, the authors found that people who lived in more segregated US Commuter Zones were less likely to agree with the World Values Survey item stating that incomes should be made more equal. The authors further found that participants who were presented with an organisation or country where economic inequality was more (versus less) segregated rated these contexts as less unequal. They also rated the economic dispersion as more fair and less concerning. Additionally, in the final two studies, the authors asked participants to rate the extent to which they would think about their and others' income in the country, and found significant indirect effects of the segregation manipulation on fairness and concern through social comparisons.

I found this a very enjoyable and thought-provoking paper that will make a valuable contribution to the field. Indeed, I have very little in the way of criticism to offer as the package of studies present a compelling and seemingly robust set of results.

I do, however, think that the theorising could be further refined. Specifically, I feel that the authors were less clear than they could have been about the different pathways through which economic segregation could affect perceptions of inequality. The first, more obvious, pathway is through the exposure to a biased set of economic cues. That is, rich people are likely to assort into social groups and institutions that mean that they are exposed to fewer poverty cues than poor people (and vice versa). The second, less obvious, pathway is by changing the distance between cues of wealth and poverty and in this way reducing the extent to which these cues are juxtaposed or easily compared. While either pathway could be operating in the archival data, the experimental work focuses on the latter path. I don't think that this is at all problematic, as the latter pathway is really quite intriguing. Instead, I think that the authors could do a better job of clearly contrasting between these pathways and acknowledging that both could be involved in the archival dataset.

Another issue that I think would benefit from further consideration is the author's focus on social comparison as the mediator of the impact of economic segregation on judgments of the fairness of and concern over the distribution of economic outcomes. The authors foreground this in the introduction, but only test it in two studies. While they find evidence that is consistent with their mediational claim, they don't consider the other variable that to me feels rather more compelling: perceptions of inequality. Across all their studies, the authors find that when economic inequality is segregated people perceive the organization or country as less unequal. Indeed, the size of this effect is generally the largest of those measured. It would seem more parsimonious to me that it is because

people see less inequality when it is segregated (even if they are presented with identical inequality cues) that they are less concerned. This possibility aligns more closely with the visual illusion presented in the introduction and would be worth more attention (as would exploratory indirect effects analysis).

The only other point that I'd raise is that I'm not aware of a South African city called Dunbar, although there is one called Durban.

Response to reviews

We are extremely grateful for the supportive, detailed, and insightful reviews made by the four anonymous reviewers. We were especially encouraged by the positivity of the reviews which highlighted the manuscript's strengths and bolstered our belief in the work's importance, robustness, and novelty. Specifically, the reviewers believed that the manuscript presents a "*strong paper*" that is "*very enjoyable and thought-provoking*" and which articulates "*an interesting narrative*." Moreover, the reviewers complimented the paper for including "*a compelling and seemingly robust set of results*" that do "*a great job*" of examining "*the same pattern from different angles*" and provide "*an excellent way to increase confidence and reduce researcher degrees of freedom*." Equally important, the reviewers lauded the paper for exploring "*an important topic*" and "*an interesting hypothesis*" which "*will make a valuable contribution to the field*."

Of course, the reviewers also brought up several helpful and constructive comments that helped us strengthen the manuscript's empirical basis and theoretical grounding. We are grateful for these suggestions and devoted a considerable amount of time addressing them by collecting more data and significantly editing and expanding the scope of the Introduction and the General Discussion. Among other things, we added three new studies to the manuscript (Studies 2b, 2c, and 3) that replicate and extend our findings and that address each and every one of the reviewers' suggestions.

Our goal is to make this paper as theoretically and empirically robust as possible and we sincerely thank you for guiding us in the revision process and for giving us the opportunity to do so. Below, we report all of the reviewers' questions and comments (in blue) and describe in

detail how these were addressed (**in black**). As a result, we believe that the manuscript is substantially stronger and more impactful.

REVIEWER COMMENTS

Reviewer #1 (Remarks to the Author):

The article “Economic segregation reduces concern about economic inequality” analyses the perception of inequality along different dimensions mainly through experimental methods by varying how different neighbourhoods/people are presented together or separately. I think the authors do a great job of conducting a set of experiments that try to get at the same pattern from different angles. However, there are two concerns, one of which can be alleviated easily by reformulating some parts of the study, while the other might be more difficult for the authors and involve additional data collection.

Theory/interpretation: It seems that the authors come from a very experiment-focused background (my apologies if this is not correct). I, on the other hand, tend to use mainly survey data, which might explain this critique: I think the experiments show interesting patterns of participant’s responses being exposed to different sets of numbers or images. However, I think this is what is primarily happening, people react to numbers and pictures. This is, per se, not bad, but I would caution a more careful interpretation of the findings. Connecting these very abstract experimental settings to large-scale social concerns about preferences for policies about inequality and residential segregation jumps at least one step. I would find the argument more convincing that starts by the likely real-world associations that participants have when completing the experiments and extrapolating what this can mean for their views about the world. E.g. what company do people think about that has extreme inequality within offices, compared to one that has inequality between offices? What real-world associations might these examples elicit and what can be learned from this? I think this study has some interesting insights, especially given that different experiments were combined, but I would be more convinced by an interpretation from the bottom and a more careful conclusion about what this means for country-wide concerns about inequality.

- 1. Reviewer 1 wondered about the real-world relevance of our findings and suggested that the paper could benefit from connecting the “*abstract experimental settings to large-scale social concerns.*” Since we share Reviewer 1’s belief in the importance of real-world experiences, we took two important steps during the revision process:**

First, we conducted a new, pre-registered study (now Study 3) in which we examined how economic segregation affects concerns about inequality in a real-world organization. In this study, we showed participants information about the actual wages in 25 different positions at a large, multinational technology company (*Amazon.com*; reported by *Glassdoor.com*—an online recruiting site where employees anonymously share information about their wages) and randomly assigned them to see the positions as economically segregated (i.e., five highest paid positions were separated from the second highest five positions, which were separated from the next highest 5 positions, and so on) or economically integrated (i.e., high- and low-paid positions were presented together, in a way that emphasizes the contrast between them). Then, after seeing all 25 positions, participants reported how concerned they are about the overall level of inequality at *Amazon* and how fair they judged it to be. Replicating our findings in the context of a real-world organization, participants were much less concerned about wage disparity between different positions at *Amazon* when these jobs were seen as segregated from each other. Thus, adding additional support for the robustness of the hypothesized effect, we found that economic segregation reduced concerns about inequality even when it came to a real and well-known corporation. This new Study 3 can be found on pages 15-17 of the revised manuscript.

In addition to collecting new data, we revised the Discussion section of the manuscript in order to explicitly state the potential limitations of generalizing from lab-based studies to large-scale social concerns. Specifically, as we now note in the revised

manuscript, economic segregation “in the wild” may interact with other factors that are not captured by the current methods and thus caution is warranted when drawing conclusions to large-scale social concerns. At the same time, we make sure to note that the analysis of the archival data in Study 1 complements the robust experimental evidence in the paper and thus highlights the real-world impact and relevance of economic segregation. Consequently, the use of a multi-method approach in which different studies complement each other’s limitations bolsters our confidence in the findings’ relevance to “the real world.” This discussion on the boundaries of generalizability can be found on page 23 of the revised manuscript.

2. Methodology: The study was conducted using MTurk. This, unfortunately, means that the authors have little control over data quality. Recent reports (e.g. <https://journals.sagepub.com/doi/10.1177/17456916221120027>) about the rate of valid responses that actually engage with an experiment in a meaningful way are disheartening. It is, therefore, possible that some or many of the patterns were produced by participants that only had passing attention to the questions and reacted, e.g., on the most recent picture or condition. This would question the validity of the findings.

The preregistration document stipulates that “Participants will be excluded for failing simple attention check questions”, which is a good start. However, regular MTurk participants should be able to spot them by now and be able to answer them properly even when trying to minimize the time spent.

Having some replication of the results with an in-person pool of experimental participants in a physical lab for whom ensuring data quality is easier would greatly alleviate my concerns that the results reflect people engaging with questions, rather than clicking through a survey as quick as possible to make money.

- 2. Reviewer 1 noted that the experimental studies rely on online samples collected through Amazon’s Mechanical Turk and suggested that the paper could benefit from alternative participant pools. Although we recruited participants from online samples due to their diversity in terms of socioeconomic status (relative to lab-based samples)—a construct that is clearly related to our research topic—we completely**

understand the reviewer’s concern. Thus, while data quality in online samples has been shown to be comparable to lab-based samples, we took Reviewer 1’s point to heart and addressed it in three critical ways. First, as part of the revision process, we conducted two new pre-registered studies (Studies 2b and 3) that sampled from Prolific Academic—an online alternative to Amazon’s Mechanical Turk that is known for its relatively high data quality. Second, as suggested by Reviewer 1, we ran a pre-registered direct replication of one of these new studies in a research lab at a university in the U.S. (Study 2d). Of note, this direct replication found almost identical results to the Prolific Academic research sample, thus alleviating concerns about data quality in the reported experiments. Finally, we made sure to include open-ended attention checks in our experimental studies (i.e., attention checks that are unlikely to be correctly answered by ‘bots’; see the available research materials on Open Science Framework) and report for each study the exact number of participants who failed these checks. We believe that the inclusion of two new studies using alternative participant pools and an additional third study that replicates the findings with participants from a behavioral research lab greatly strengthens the evidential basis of the manuscript and alleviates concerns about data quality.

Reviewer #2 (Remarks to the Author):

The article builds an interesting narrative connecting economic segregation and the perception of inequality. Its main objective is to test the hypothesis that individuals from segregated economies are less concerned about economic inequality. To achieve this, the authors employed a combination of one quantitative analysis and five qualitative experiments, using a hypothetical company, city, satellite images from a highly unequal country, and archival datasets. Based on their findings, the authors conclude that individuals from less segregated environments exhibit

greater concern about inequality than their more segregated counterparts. They propose that the juxtaposition of inequality in less segregated scenarios, which disappears in segregated scenarios, is the main force driving a higher concern about inequality.

Overall, this is a strong paper potentially suitable for Nature Communications-- however, I would love to see implemented the issues discussed below before a final assessment is made.

General assessment of the work

The authors should provide additional justification for their claim that economic segregation eliminates the juxtaposition of inequality for both poor and rich individuals. A notable concern is the lack of references to support the argument that economically-segregated societies display less concern about inequality due to the inhibiting effect of segregation on the comparison between the rich and the poor in various aspects such as work, education, and housing. The central hypothesis and conclusion of the manuscript rely on this assumption, which has yet to be adequately supported in the introduction and discussion sections. For instance, the example provided (lines 75-77) of an expensive meal and a homeless person is used to support the hypothesis that the rich and the poor are not juxtaposed in segregated societies. However, it can be argued that in highly unequal countries, middle-upper-class restaurants do have street vendors entering, thus challenging this example. Similarly, in the discussion (line 358), the same assumption is made without being substantiated with references.

- 3. Reviewer 2 raises an interesting question regarding the effect of economic segregation across different countries. Specifically, the reviewer wondered whether more segregated societies display less concern about inequality than economically integrated societies. Unfortunately, since our paper is the first to ever examine the effect of segregation on concerns about inequality, available data for analysis on this topic is lacking. Since we are unaware of any cross-country dataset of economic segregation, we couldn't examine the global effect of segregation and focused instead on its effect within the U.S. Nevertheless, we completely agree with Reviewer 2 that more research is needed to comprehensively compare the level of segregation across different countries, and now make sure to note this as a potentially rich avenue for future research in the Discussion section of the revised manuscript (page 23).**

4. **Reviewer 2 wondered about the example regarding the importance of segregation for attitudes about inequality. Although (as correctly noted by Reviewer 2), street vendors do sometimes enter restaurants in highly unequal countries, we suggest that this is less likely to occur in unequal countries that are also highly *segregated*. That is, holding the level of overall inequality constant, cross-class interactions occur more frequently in economically integrated areas than in segregated areas. Consequently, we argue that the frequency of such cross-class interactions in integrated areas may increase concerns about inequality. Nevertheless, upon considering Reviewer 2's comments, we edited the example in the manuscript to clarify that segregation is a situational characteristic that is independent from an area's level of inequality. Of course, since we highly value clarity in scientific reporting, we would be happy to consider other potential examples that could help illustrate the hypothesized effect of economic segregation on concerns about inequality.**

While the authors effectively demonstrate that the absence of juxtaposition between economic classes diminishes concerns about inequality, it remains unclear whether economic segregation itself is responsible for this reduction. The study convincingly supports the notion that when individuals from different economic backgrounds are physically separated and lack direct exposure to one another, their concerns about inequality may diminish. However, the specific causal relationship between economic segregation and complete lack of juxtaposition requires further clarification. The manuscript would benefit from additional evidence or theoretical reasoning to establish a clearer link between these two concepts.

5. **Reviewer 2 also noted that the paper could benefit from bolstering the theoretical link between segregation and the juxtaposition between people of different economic means. We completely agree with this suggestion and have done exactly that. First, we added several references to the paper that discuss how segregation reduces the**

juxtaposition between people of different means. As we now note in the revised manuscript, while economic segregation doesn't completely eliminate exposure to people of different means, it substantially increases class isolation and reduces the frequency of cross-class interactions. For instance, as wealthy people increasingly segregate into homogeneously affluent neighborhoods, they become less likely to encounter middle- and lower-class others (and the disproportionately disadvantaged conditions that they face) which translates into decreased exposure to environments where people can experience the spatial juxtaposition between the rich and the poor. Second, in revising the manuscript, we articulated that the (lack of) juxtaposition between wealth and poverty is, by definition, an inherent to economic segregation. By separating people based on their economic resources, segregation reduces, by definition, the juxtaposition between them. Similar to the Mach Bands example that is discussed in the manuscript, as the physical distance between people of different means increases, the contrast between them decreases. This does not mean that economic segregation completely eliminates the juxtaposition between the rich and the poor. Rather, as we now explicitly state in the manuscript, economic segregation reduces the frequency and salience of this juxtaposition, reducing (but not completely eliminating) the contrast between people of different means. We believe that these revisions (which are found in the second paragraph of the revised Introduction) help clarify this relationship and thank Reviewer 2 for this comment.

Major comments

A concern arises regarding Study 1, which utilizes data on economic segregation that is primarily based on residential segregation. This limited focus on residential segregation paints an incomplete picture of overall segregation encompassing various domains such as work, school,

church, and others, which the authors intend to incorporate. To strengthen their argument, the authors should provide further justification for how residential segregation adequately represents economic segregation across society, as stated in the conclusion (lines 377-380).

6. Reviewer 2 wondered why the archival dataset that was used in Study 1 focuses on *residential* economic segregation. While we agree that examining the hypothesized effect using additional operationalizations of economic segregation can be fruitful for future research, there are two reasons why we focused on *residential* economic segregation. First, as we now mention in the manuscript, where people live corresponds highly with where they work, socialize, and send their children to school, suggesting that residential segregation closely approximates other types of segregation. Indeed, many of the papers that we now cite in the manuscript rely on *residential* economic segregation data. Second, we focused on residential economic segregation due to the unavailability of other openly available large-scale datasets of economic segregation. Thus, given the theoretical relevance of *residential* economic segregation, and due to the lack of comprehensive alternative datasets, we focused on residential segregation. Nevertheless, when examining the causal effect of segregation on concerns about inequality (Studies 2-5), we made sure to operationalize economic segregation in various ways, focusing on economic segregation in the workplace as well as in people's homes. We thank Reviewer 2 for giving us the opportunity to explain and clarify this empirical choice!

Minor comments

- I 227,228: Authors write that participants believed that wage inequality requires more attention in the Economic Segregation condition than in the Economic Integration condition but it should be the other way around.

7. Thank you for noting this. We corrected this error!

- 1251 and 1289: Manipulation check to rate inequality for each city/neighborhood shows less perceived inequality in the economic segregation scenario. Authors should comment on how this does not bias the overall perception of inequality in the country.

8. As noted in the manuscript, the manipulation check examines perceptions of inequality within each city, neighborhood, or office. Given that we manipulate whether people are segregated by income, inequality should be higher within each specific locale (in fact, if this was not the case, we wouldn't be able to demonstrate that the manipulation affected what it is supposed to affect). That being said, participants' perceptions of the *overall* level of inequality in the country/company (i.e., across several cities, neighborhoods, or offices) are important and require extra attention. Indeed, as discussed in detail in point #9 (see below), in revising the manuscript we have done exactly that. Specifically, we conducted a within-participant study in which participants' perceptions of the overall inequality are held constant, showing that economic segregation in and of itself can lead to lower concerns about inequality. Thus, while the differences in perceived local inequality in the manipulation check are *by design*, we discuss in the revised manuscript how differences in the perceived *global* inequality (i.e., across several locales) are a sufficient but not necessary alternative pathway for the effect of segregation on attitudes about inequality.

Reviewer #3 (Remarks to the Author):

The authors present one correlational archival study showing a robust negative association between local economic segregation and concern about economic inequality, four experiments using hypothetical vignettes to show people have greater concern over inequality in less

economically segregated companies and cities, and one additional experiment finding similar results using aerial photos of unequal neighborhoods in South Africa.

There is a lot to like about this paper. It addresses an important topic with an interesting hypothesis. Study 1's analysis of testing various sets of variable combinations is an excellent way to increase confidence and reduce researcher degrees of freedom in this study. And I especially liked the Mach band analogy.

At the same time, I feel the paper is significantly undermined by how the authors misrepresent, or perhaps simply misframe their findings. Over and over again throughout the manuscript, the authors emphasize that what differentiates this work from the prior literature is that their effects occur "holding constant what people know about inequality" and "even when people are fully aware of otherwise identical levels of economic disparities between the rich and the poor" allowing them "to isolate the direct effect of segregation on attitudes about inequality independent of any misperceptions." The central claim, as I understand it, is that prior work has shown that people are more bothered about inequality when they perceive more of it. But, even when people perceive the same amount of inequality, their concern over it will be directly affected by how well-integrated or segregated those inequalities are. My issue is that in every one of their experiments, people in the economically integrated condition perceive there to be more inequality than those in the economically segregated condition. Indeed, taking a quick peek at the Study 3 data (I appreciate the authors making all their data and materials available and easy to navigate), the difference in the perceived level of inequality is larger between the economically segregated and integrated conditions than it is between the Gini conditions, in which inequality actually differed. Moreover, the perception of inequality fully mediates the effect of the economic segregation manipulation on the concern about inequality, on fairness, and on support for action.

What prior work, like that of Sands, shows is that when the salience of inequality is raised (often via local instances of inequality), people perceive more inequality, and their concern affected. This work shows something similar, but does so by finding that by presenting people with different wages or levels of wealth in different local groupings, it changes how much global inequality is perceived, and in turn people's concern is affected.

Now this is an interesting finding in its own right, but it seems like it is an importantly different one from the one the authors present in the paper. To match the conclusions that the authors draw in the paper, I would think that the authors would want to show that the concern dependent measures are higher in economically integrated conditions even when participants recognize that the inequality is objectively the same. Take the lovely Mach bands example that the authors use. What makes that such a compelling demonstration is that we know that the bands are the same colors when presented together and apart, but fall for the illusion nonetheless. In these studies, however, people don't think that the inequality is the same. If the authors wish to test the conclusion they espouse, they could construct a study in which, just like people would report explicitly understanding the Mach Band to be the same colors in the two presentations, people explicitly recognize the global inequality to be the same. For instance, they could first show the full group of wages, and then show participants how this same population can be grouped in different ways. One would assume that people would explicitly

report the inequality be the same, but one could test whether different levels of concern emerge depending on how segregated or integrated the groupings ended up being.

Alternatively, the authors could change the framing of their paper to downplay the admittedly central claim that different levels of concerns emerge “even when people are fully aware of otherwise identical levels of economic disparities”. Instead they could focus on the more cognitive process of how the different local grouping affects perceptions of global inequality. Although I’m not sure that would be as interesting as how the authors are currently framing things.

9. Reviewer 3 noted that economic segregation in Studies 2 and 3 affected perceptions of the scope of inequality and wondered about the role of such perceived differences in the effect of segregation. This point was also raised by Reviewer 4 (see below), who suggested that two separate pathways may account for the hypothesized effect of segregation on concerns about inequality: differences in perceived inequality and differences in economic comparisons. We completely agree with this suggestion and have made sure to note so in the revised manuscript. Specifically, as we now discuss in the revised Introduction and Discussion sections, the proposed mechanism for the hypothesized effect (i.e., economic comparisons) does not rule out the existence of other potential mechanisms and the two pathways may independently account for the robust effect of segregation on attitudes about inequality. Thus, rather than ruling out the role of perceived inequality, our findings rules in the role of reduced comparisons as an additional and independent process. Put differently, our findings suggest that each proposed pathway—perceived inequality and economic comparisons—may be sufficient for explaining the effect of economic segregation that our paper is first to document. Nevertheless, as we now discuss in the revised manuscript, these two distinct pathways may still interact with each other, and future research could focus on the nature of such potential interactions. The discussion of the contributing role of

perceived inequality can be found on page 5 of the revised Introduction and page 24 of the revised Discussion section.

Importantly, the two pathways for the effect of economic segregation—perceived inequality and economic comparisons—are empirically distinguishable. Indeed, while Reviewer 3 correctly noted that the effect of segregation in Study 3 can be explained by differences in perceived inequality, an analysis of Study 2 (now Study 2a) found that the effect of segregation is exhibited above and beyond the effect of perceived inequality. Thus, while differences in perceptions may contribute to the hypothesized effect of segregation, this effect persists even when controlling for such differences, suggesting the existence of other contributing factors.

Nevertheless, we agree that more data is needed to distinguish between the two pathways for the hypothesized effect of economic segregation. Thus, in revising the paper, we conducted two new studies (a pre-registered online study and a replication in a behavioral research lab) that manipulate economic segregation while keeping the level of perceived inequality constant. Directly following Reviewer 3's suggestion, we ran two within-participants studies in which participants saw a group of 20 employees across four offices and *“how this same population can be grouped in different ways”*—an economic segregated way and an economically integrated way. After seeing each potential grouping, participants reported their concerns about the level of inequality in the organization that they saw. Not surprisingly, this empirical design made it clear to participants that the overall level of inequality was the same across the different groupings. Yet, even though they realized that the different groupings share the same

level of overall inequality, participants were much less concerned about it when the employees were segregated based on pay. That is, while perceived inequality affects attitudes about it, these two new studies reveal that economic segregation reduces concerns about inequality independent of such distorted perceptions, making it clear that economic comparisons may independently play an important role in this process. Thus, the hypothesized effect of segregation on attitudes about inequality likely occurs through two distinct pathways: by distorting perceptions of inequality and by weakening people’s tendency to engage in economic comparisons. These two new studies (Studies 2b and 2c) are found on pages 11-13 of the revised manuscript.

Minor points:

In the discussion, the authors write “Notably, since economic segregation reduces the visibility of “the other” for both the rich and the poor, its effects were not moderated by income.” I might have missed the discussion of a lack of moderation by income. Was that for the first study? The subsequent studies seem less relevant since the income of the participant has nothing to do with the people or situation they are rating—which is the important question. Given Sands’ findings that the rich and poor respond differently in response to exposure to inequality, it would be very interesting to see whether similar effects are seen here. That said, one would want to make the ratings relevant to the participants, so that they had “skin in the game” as rich and poor people do in real life.

10. Reviewer 3 wondered about potential moderation by income and noted that people’s own income may be less relevant when they consider situations as external observers. Indeed, although we did not find any evidence of moderation by income, this does not mean that the effect of segregation is homogenous for people across the economic spectrum and we thus cannot rule out the existence of such moderation. Moreover, as we now discuss in the manuscript, even if economic segregation affects people across the economic spectrum, it is unclear whether concerns about inequality translate to

similar behaviors among people of different means. Therefore, following Reviewer 3's advice, we made sure to note in the revised manuscript that the moderating effect of inequality remains an open question for future research and that our discussion of this topic remains speculative (pages 23-24).

For Study 4, it would be great to include examples of the stimuli so that readers could get a sense for themselves how differently economically integrated images feel compared to segregated ones.

11. We completely agree. We made sure to include all the stimuli from the experimental studies in the online materials on the Open Science Framework. We hope that making our experimental materials openly available will allow researchers to replicate and extend our findings.

Reviewer #4 (Remarks to the Author):

In this paper, the authors report the results of 6 studies (1 archival study and 4 experiments, with an additional direct replication) that aimed to explore the impact of economic segregation on people's perceptions of, and concern about, economic inequality. In line with their expectations, the authors found that people who lived in more segregated US Commuter Zones were less likely to agree with the World Values Survey item stating that incomes should be made more equal. The authors further found that participants who were presented with an organisation or country where economic inequality was more (versus less) segregated rated these contexts as less unequal. They also rated the economic dispersion as more fair and less concerning. Additionally, in the final two studies, the authors asked participants to rate the extent to which they would think about their and others' income in the country, and found significant indirect effects of the segregation manipulation on fairness and concern through social comparisons.

I found this a very enjoyable and thought-provoking paper that will make a valuable contribution to the field. Indeed, I have very little in the way of criticism to offer as the package of studies present a compelling and seemingly robust set of results.

I do, however, think that the theorising could be further refined. Specifically, I feel that the authors were less clear than they could have been about the different pathways through which economic segregation could affect perceptions of inequality. The first, more obvious, pathway is through the exposure to a biased set of economic cues. That is, rich people are likely to assort into social groups and institutions that mean that they are exposed to fewer poverty cues than poor people (and vice versa). The second, less obvious, pathway is by changing the distance

between cues of wealth and poverty and in this way reducing the extent to which these cues are juxtaposed or easily compared. While either pathway could be operating in the archival data, the experimental work focuses on the latter path. I don't think that this is at all problematic, as the latter pathway is really quite intriguing. Instead, I think that the authors could do a better job of clearly contrasting between these pathways and acknowledging that both could be involved in the archival dataset.

12. As noted in detail above (in point #9), we addressed this comment in three ways: by incorporating into the paper's theory the two distinct pathways idea that Reviewer 4 raises, by further analyzing the data to show that the effect of economic segregation is apparent even when controlling for perceived economic inequality, and (most importantly) by conducting two new studies with both an online sample and a sample of a research lab that explicitly hold the level of perceived inequality constant across the two experimental conditions. As we discuss above, these edits and additions substantially bolster the paper, make it more theoretically and empirically robust.

The only other point that I'd raise is that I'm not aware of a South African city called Dunbar, although there is one called Durban.

13. Thank you. We corrected this type! We would like to take this opportunity to reiterate our gratitude to the editorial team for paying close attention to the manuscript, which we see as a sign of its potential. Thank you!

Reviewers' Comments:

Reviewer #2:

Remarks to the Author:

I appreciate the authors' thorough responses to my comments, and I must acknowledge that the paper has significantly improved and addressed most of my concerns regarding the relationship between economic segregation and the juxtaposition of individuals from different economic classes (referred to as "global segregation" by the authors). However, I would like to suggest that the authors consider the possibility that economic segregation alone may not be the sole cause of social juxtaposition.

The paper now provides substantial evidence demonstrating the relationship between economic segregation and social juxtaposition (global segregation), which has alleviated many of my concerns. Yet, I still believe it's important to recognize that global segregation (social juxtaposition) is a multi-dimensional phenomenon, with economic segregation being a significant component, as convincingly argued in the revised version. However, it is not the sole driver of this phenomenon.

For instance, factors such as exposure to news about economically disadvantaged areas, daily interactions with individuals from different social classes (common in many cities), family dynamics, and interactions with employees, among others, all contribute to the complex web of global segregation. While some of these factors might not be consciously perceived and might not significantly influence the perception of societal inequality, they still play a role in shaping the dynamics of global segregation.

I believe that the manuscript could provide valuable insights for future research if it includes a brief discussion in the conclusion section regarding the concept of global segregation and its relationship with social juxtaposition. This would enhance the comprehensiveness of the paper.

With this minor addition, the manuscript would be well-prepared for publication.

Reviewer #3:

Remarks to the Author:

The authors have addressed my primary concern about whether these effects occur even when people believe that the overall level of inequality has remained the same. Most compellingly, they have presented two new studies testing this exact question. I am satisfied with this revision and appreciate the efforts the authors have gone to to address my comments.

Response to reviews

We are thankful to the editor and two reviewers for their dedication to our paper and for giving us an opportunity to address any remaining issues. We were glad to learn that the reviewers believe that the revision process has “significantly improved” our paper, and we look forward to the hopefully engaging and constructive academic exchange that its publication will inspire. To address the remaining issues in this final round of review, we have done the following:

Reviewer #2 (Remarks to the Author):

I appreciate the authors' thorough responses to my comments, and I must acknowledge that the paper has significantly improved and addressed most of my concerns regarding the relationship between economic segregation and the juxtaposition of individuals from different economic classes (referred to as "global segregation" by the authors). However, I would like to suggest that the authors consider the possibility that economic segregation alone may not be the sole cause of social juxtaposition.

The paper now provides substantial evidence demonstrating the relationship between economic segregation and social juxtaposition (global segregation), which has alleviated many of my concerns. Yet, I still believe it's important to recognize that global segregation (social juxtaposition) is a multi-dimensional phenomenon, with economic segregation being a significant component, as convincingly argued in the revised version. However, it is not the sole driver of this phenomenon.

For instance, factors such as exposure to news about economically disadvantaged areas, daily interactions with individuals from different social classes (common in many cities), family dynamics, and interactions with employees, among others, all contribute to the complex web of global segregation. While some of these factors might not be consciously perceived and might not significantly influence the perception of societal inequality, they still play a role in shaping the dynamics of global segregation.

I believe that the manuscript could provide valuable insights for future research if it includes a brief discussion in the conclusion section regarding the concept of global segregation and its relationship with social juxtaposition. This would enhance the comprehensiveness of the paper.

With this minor addition, the manuscript would be well-prepared for publication.

Thank you very much for these comments. As suggested, we added a paragraph to the Discussion section of the revised manuscript in which we discuss the multi-faceted nature of perceived juxtaposition and suggest future avenues for research on the topic. Specifically, as we now note in the manuscript, various other factors beyond economic segregation may potentially weaken concerns about inequality by reducing the juxtaposition of wealth and poverty, including “a lack of exposure to news articles about economically varied regions, working with an economically homogenous customer base, attending private (vs. public) institutions of higher education, using private (vs. public) means of transportation, and so forth.” This new addition to the Discussion section can be found on page 23 of the revised manuscript.

Reviewer #3 (Remarks to the Author):

The authors have addressed my primary concern about whether these effects occur even when people believe that the overall level of inequality has remained the same. Most compellingly, they have presented two new studies testing this exact question. I am satisfied with this revision and appreciate the efforts the authors have gone to to address my comments.

Thank you very much for these comments. We agree that the two new studies have substantially contributed to both the theoretical and empirical bases of the paper, and look forward to seeing the academic discussions our paper may bring about.

Reviewers' Comments:

Reviewer #2:

Remarks to the Author:

I appreciate the authors' thorough and successful response to my comments, and I recommend this paper for publication.

Reviewer #5:

None